# Coupled otolith and foraminifera oxygen and carbon stable isotopes evidence paleoceanographic changes and fish metabolic responses

Konstantina Agiadi[1], Iuliana Vasiliev[2], Geanina Butiseacă[3], George Kontakiotis[4], Danae Thivaiou[5], Evangelia Besiou[4], Stergios Zarkogiannis[6], Efterpi Koskeridou[4], Assimina Antonarakou[4], Andreas Mulch[2,7]

[1] Department of Geology, University of Vienna, Josef-Holaubek-Platz 2 (UZA II), 1090, Vienna, Austria
[2] Senckenberg Biodiversity and Climate Research Centre, Senckenberganlage 25, 60325, Frankfurt am Main, Germany
[3] Department of Geosciences, Institute of Archaeological Sciences, Palaeoanthropology, Eberhard Karls Universität Tübingen, Rümelinstraße 23, 72070, Tübingen, Germany
[4] Department of Historical Geology and Palaeontology, Faculty of Geology and Geoenvironment, National and Kapodistrian University of Athens, Panepistimiopolis, 15784, Athens, Greece
[5] Natural History Museum of Basel, Augustinergasse 2, 4051 Basel, Switzerland
[6] Department of Earth Sciences, University of Oxford, South Parks Road, Oxford, OX1 3AN, UK
[7] Goethe University Frankfurt, Institute of Geosciences, Altenhöferallee 1, 60438 Frankfurt, Germany

*Correspondence to*: Konstantina Agiadi (konstantina.agiadi@univie.ac.at)

**Abstract.** Capturing the mechanisms leading to the local extirpation of a species in deep-time is a challenge. Combining stable oxygen and carbon isotopic analyses on benthic and planktonic foraminifera and the otoliths of pelagic and benthic fish species, we reveal here the paleoceanographic regime shifts that took place in the Eastern Mediterranean from 7.2 to 6.5 Ma, in the precursor phase to the Messinian Salinity Crisis, and discuss the fish response to these events. The step-wise restriction of the Mediterranean–Atlantic gateway impacted the metabolism of fishes in the Mediterranean, particularly those dwelling in the lower, deeper part of the water column. An important shift in the Mediterranean paleoceanographic conditions took place between 6.951 and 6.882 Ma, from predominantly temperature to salinity control, which was probably related to stratification of the water column. A regime shift at 6.814 Ma due to changes in the influx, source and/or preservation of organic matter led to pelagic–benthic decoupling of the fish fauna. At that time, the oxygen isotopic composition of benthic fish otoliths reflects higher salinity of the lower part of the water column that is accompanied by rapid fluctuation in the carbon isotopic composition (a proxy of metabolic rate), ultimately leading to the local extirpation of the benthic species. Overall, our results confirm that otolith stable oxygen and carbon isotope ratios are reliable proxies for paleoceanographic studies and, when combined with those of foraminifera, can reveal life history changes and migration patterns of teleost fishes in deep time.

## 1 Introduction

The modern trajectory of climate change makes the need for an improved understanding of the long-term resilience of marine fishes and their populations urgent. Over time, in the face of environmental change, the response of organisms shifts

from phenotypic plasticity to physiological changes, evolutionary adaptation or extinction (Donelson et al., 2019). To this end, deep-time studies can contribute greatly, because they provide the necessary baselines, potential past analogues and long time series that cover the onset and recovery from such events (Dietl et al., 2015; Leonhard and Agiadi, 2023). The paleoceanographic changes in the Mediterranean during the Messinian (7.25–5.33 Ma) and their impact on higher organisms can offer unique insights into the resilience of marine ecosystems under extreme environmental conditions. Late Miocene cooling (Herbert et al., 2016) superimposed on the gradual restriction of the marine gateway with the Atlantic starting at the Tortonian/Messinian boundary (Krijgsman et al., 1999; Flecker et al., 2015) led to profound environmental changes in the Mediterranean, culminating in the Messinian Salinity Crisis (MSC; 5.97–5.33 Ma; Hsü et al., 1973). Basin's restriction resulted in high-amplitude shifts in sea surface temperature (SST) and salinity (SSS), and episodic water-column stratification and dysoxia on the sea bottom, even before the MSC (e.g., Moissette et al., 2018; Vasiliev et al., 2019; Sabino et al., 2020; Bulian et al., 2021; Zachariasse et al., 2021; Kontakiotis et al., 2022; Mancini et al., 2024). However, the impact of these events on higher organisms such as fishes remains unclear. In the Early Messinian, the presence of new endemics (Girone et al., 2010) and Paratethyan-affinity fish species (Schwarzhans et al., 2020) in the Mediterranean has been attributed to the changing paleoceanographic conditions. This study presents a new approach on how climate affected marine life in the geological past, based on oxygen ($\delta^{18}$O) and carbon ($\delta^{13}$C) isotopic analyses of otoliths, aiming to test the hypothesis that the pre-MSC amplitude of changes in important paleoceanographic parameters (SST, SSS, organic matter source and primary productivity) had a negative impact on the physiology of marine fishes, and to identify the principal parameter controlling fish populations.

Otoliths are accretionary, mostly aragonitic structures in the inner ear of teleost fishes, and they are metabolically inert, making them excellent paleoceanographic archives (Campana, 1999). The oxygen isotope ratio in otoliths ($\delta^{18}O_{oto}$) is a function of temperature and the $\delta^{18}$O values of ambient water, which also depends on salinity: $\delta^{18}O_{oto}$ is not affected by somatic growth or the otolith precipitation rate, but varies with species-specific lifestyle patterns (Kalish, 1991). The carbon isotopic composition of fish otoliths ($\delta^{13}C_{oto}$) reflects the $\delta^{13}$C values of the fish diet and the dissolved inorganic carbon (DIC) in ambient water (Kalish, 1991; Solomon et al., 2006; Trueman et al., 2016; Chung et al., 2019a). $\delta^{13}C_{oto}$ is a proxy of fish metabolic rate (Wurster and Patterson, 2003; Solomon et al., 2006; Trueman et al., 2016; Chung et al., 2019a, b; Martino et al., 2020; Smoliński et al., 2021; Trueman et al., 2023; Jones et al., 2023), reflecting the amount of energy the fish uses to live and grow, which impacts its behavior and resilience in the face of environmental change (Gauldie, 1996; Chung et al., 2019a). At the evolutionary level, higher metabolic rates in a population are generally expected to lead to higher genetic mutation rates (Trueman et al., 2016). In addition, $\delta^{13}C_{oto}$ is linked to oxygen consumption through the metabolic oxidation of dietary carbon (Chung et al., 2019b), while high metabolic rate results in higher oxygen consumption, higher activity, higher respiration and greater carbon export (Chung et al., 2019a). Respiratory (or metabolic) carbon has a 15‰ lower $\delta^{13}$C than DIC because of preferential incorporation of $^{12}$C during photosynthesis (Kroopnick, 1985).

Here, we investigate the link between fish physiology and paleoceanographic change due to the restriction of the marine gateways connecting the Mediterranean Sea with the Atlantic Ocean in the Late Miocene. We first detect regime shifts by

analyzing time-series of independently obtained proxies. Then, we examine the fish response by comparing $\delta^{18}O_{oto}$ and $\delta^{13}C_{oto}$ time-series obtained for two very common zooplanktivorous species, the pelagic *Bregmaceros albyi* and the benthic *Lesueurigobius friesii*, from the Eastern Mediterranean for the interval 7.2–6.5 Ma.

## 2 Material and Methods

### 2.1 Material and sampling

Forty-seven sediment samples were collected from the ~25-m–thick pre-MSC Messinian laminated (sapropelic) and homogeneous marls succession of Agios Myron (Fig. 1; N 35°23'35.90", E 25°12'67.91", Crete Island, Eastern Mediterranean). The age model for the Agios Myron section has been published by Zachariasse et al. (2021). The Agios Myron section presents lithological precessional cycles (confirmed by the vanadium concentrations in the bulk sediment and spectral analysis), which have been orbitally tuned through correlation with the Metochia section (Hilgen et al. 1995; 1997; Krijgsman et al. 1995; 1999), and anchoring in planktonic foraminifera bioevents and ash layers (Zachariasse et al. 2021). The horizons where each sample was taken, along with their lithology, position along the section, age, and sample number have been provided in the supplementary data. We sampled approximately 1 kg of sediment from each analyzed horizon, including both sapropels and calcareous marls, and we used plain water to break down the sediment.

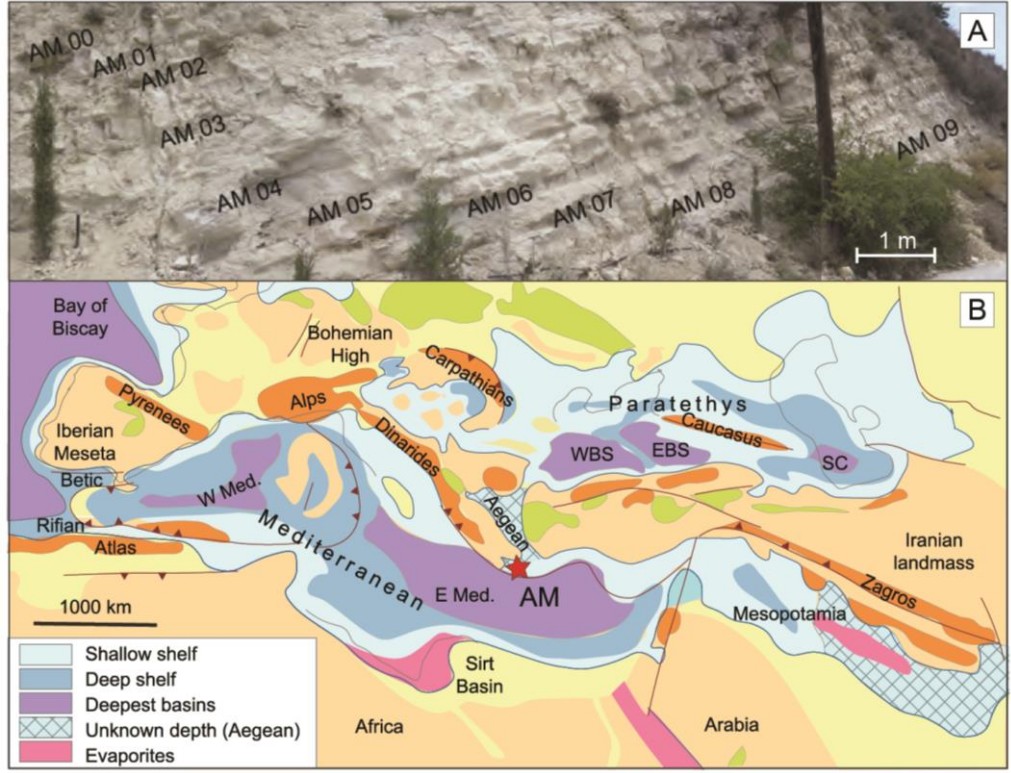

**Figure 1: Study section. A) Photograph of the section showing the lithological alternations and sampling approach. B) Paleogeography map for the Mediterranean-Paratethys domain during late Miocene (Ilyina et al., 2004). Note the paleolatitudinal location and their approximate position within the late Miocene Mediterranean basin for Agios Myron (AM). W Med denotes western Mediterranean, while E Med refers to eastern Mediterranean.**

The samples were wet-sieved using various mesh sizes (minimum 63 μm). Up to five individuals of the shallow-dwelling planktonic foraminifer species *Globigerinoides obliquus* and the benthic foraminifer *Cibicides (pseudo)ungerianus* were picked from the 250–300 μm sieve fraction of the same samples, according to standard practices (Seidenkrantz et al., 2000; Elderfield et al., 2002). The otoliths were handpicked from the sieved sediment residues and identified to species level.

From the picked specimens, the two target species, *Bregmaceros albyi* and *Lesueurigobius friesii*, were identified in 20 and 14 samples, respectively. Single otoliths from each level (and duplicates from samples AM05 and AM20C) were measured (L: length and W: width), cleaned and analyzed. We selected well-preserved otoliths, making sure that all their morphological characteristics were still present and that there were no particular coloration and/or signs of bioerosion or encrustation that may point toward extensive diagenetic alteration (Agiadi et al., 2022). Only otoliths of adult individuals

were analyzed to avoid differences in species-specific vital effects that affect $\delta^{18}O_{oto}$ values (Darnaude et al., 2014).

## 2.2 Ecological information about the selected fish species

The two fish species, whose otoliths were analyzed, were selected because of: a) their great abundance and frequency in Neogene and Quaternary Mediterranean marine sediments, which would render their isotopic compositions useful as paleoceanographic proxies, and b) their well-established ecology, allowing more accurate interpretation of isotopic analyses

results. *Bregmaceros albyi* (Sauvage, 1880) is one of the most common extinct fish species in the Neogene and Quaternary Mediterranean, found both as articulated skeletons and otoliths in great abundances (Landini and Sorbini, 2005; Agiadi and Karakitsios, 2012; Agiadi et al., 2013). *Bregmaceros* spp. are small pelagic fishes with 14 extant species distributed in the Atlantic, Indian and Pacific Oceans and mostly occupying the euphotic zone (FishBase, 2024). Even though *Bregmaceros* sp. was still present in the Eastern Mediterranean until the Late Pleistocene (Cornée et al., 2019), *B. nectabanus* Whitley,

1941 is the only representative currently inhabiting the basin, and it is considered a non-indigenous species (Agiadi and Albano, 2020). *Bregmaceros nectabanus* has been reported up to 350 m depth, but is also able to migrate temporarily into dysoxic waters (FishBase, 2024). *Bregmaceros* spp. feed on zooplanktonic invertebrates, particularly copepods, and some species also on phytoplankton (FishBase, 2024).

*Lesueurigobius friesii* (Malm, 1874) is a benthic, subtropical marine fish living at depths between 10–130 m in the Eastern

Atlantic and the Mediterranean Sea, and feeding mainly on polychaetes, small crustaceans and mollusks (FishBase, 2024). Today, it has a preferred temperature between 7.2 and 18.4 °C, with a mean of 10.4 °C (AquaMaps: predicted range maps for aquatic species). Its presence in the Mediterranean fossil record goes back to the late Tortonian (Agiadi et al., 2017) and it is common in shallow-waters until the present (Agiadi et al., 2019; Agiadi and Albano, 2020). As a benthic species, *L. friesii* is site-attached, and therefore its isotopic composition is expected to closely reflect the local conditions on the sea floor

(Mirasole et al., 2017).

## 2.3 Oxygen and carbon isotopic analyses on otoliths and planktonic foraminifera

The otoliths and the foraminifera specimens were sonicated in methanol for about 10 s to remove clay particles adhering to the tests and rinsed at least five times in ultraclean water. Randomly selected specimens of foraminifera and otoliths were observed using a Jeol JSM 6360 scanning electron microscope or a stereoscope, respectively, to confirm their good preservation state, following the established practices in the field (Antonarakou et al., 2019; Agiadi et al., 2022; Lopes et al., 2022). Complete and intact otoliths were crushed and homogenized, and the results have been interpreted to reflect whole-life conditions. Generally, otoliths seem to suffer little from diagenetic alteration, mostly maintaining their microstructure, chemistry and mineralogy. In fact, Dufour et al. (2000) discovered in their study of Pliocene otoliths from the Mediterranean that most specimens remained aragonitc, while pyrite appeared in only a few otoliths. Moreover, the isotopic composition of otoliths depends on the taxon rather than geologic age (Dufour et al., 2000). Indeed, any diagenetic alteration is focused mostly on the outer part of the otolith (Cook et al., 2015).

Stable carbon and oxygen isotopic analyses were performed using a Thermo Scientific MAT 253 Plus mass spectrometer coupled to a GasBench II, in continuous flow mode, using a thermostated sample tray and a GC PAL autosampler at the Goethe Universität – Senckenberg BiK-F Joint Stable Isotope Facility in Frankfurt am Main, Germany. Each sample (between ~80 and ~110 μg) was weighed and put in a 12-ml glass vial. The airtight glass vials were placed into a Jumo iTRON 16 temperature controller of the A200S autosampler of the Finnigan™ GasBench II from Thermo Scientific, which is attached to the mass spectrometer. The autosampler allows automated isotope ratio determination of small $CO_2$ gas samples. To displace the atmospheric gas in the glass vials, each sample was flushed with 1.2 bar He (purity > 99.999 %) for 300 seconds. After this flush-fill viscous water-free 98 g/mol orthophosphoric acid ($H_3PO_4$) was injected with a syringe through the septum into each vial. $CO_2$ gas, phosphates of calcium and water will be formed by carbonate reacting with the acid. The vials were heated the temperature controller to a constant temperature of 72 °C to speed up the reaction between the carbonates and $H_3PO_4$, and to prevent the crystallization of the acid at room temperature. Analytical precision was 0.08‰ for $\delta^{18}O$ and 0.06‰ for $\delta^{13}C$. In both cases, we analyzed replicates of 10% of the data reveal to confirm reproducibility of the results; natural sample variability was better than 0.1‰. Results are reported against the Vienna Pee Dee Belemnite (VPDB) standard using the $\delta$ notation and expressed in per mille (‰).

## 2.4 Additional biogeochemical data used for interpretation

The results of the otoliths isotopic analyses were additionally interpreted considering: a) the $\delta^{18}O$ and $\delta^{13}C$ values of *C. (pseudo)ungerianus* (Zachariasse et al., 2021); b) paleodepth estimates based on foraminifera (Zachariasse et al., 2021); c) Tetra Ether Index ($TEX^H_{86}$)-derived SST and $TEX_{86}^{H}$-$\delta^{18}O_{G.obliquus}$-based SSS estimates (Kontakiotis et al., 2022); d) the ratio of isoprenoidal glycerol dialkyl glycerol tetraethers 2 and 3 (isoGDGT2/isoGDGT3) (Kontakiotis et al., 2022), which increases with isoGDGTs input from subsurface-dwelling archaea (Schouten et al., 2002); and e) BIT (branched and isoprenoid tetraether) index values which reflect the source of organic matter (Butiseacă et al., 2022).

## 2.5 Statistical analyses

In order to test for the response of the fishes to significant changes in paleoceanographic parameters, we used the Sequential
T-test Analysis of Regime-Shifts (STARS) algorithm (Rodionov, 2004) to identify regime shifts based on the organic
biomarker and foraminifera isotopic data. Regime shifts are abrupt changes between two natural states of climate or of an
ecosystem, where each state can have its own internal variability (Kerr, 1992; Scheffer, 2020). The STARS algorithm allows
detecting regime shifts robustly even at the ends of a time series (Rodionov, 2004). Because the STARS algorithm can only
be applied to variables that have a normal distribution, we first tested for normality using the Shapiro-Wilk test. For those
variables that did not have a normal distribution (here, $\delta^{13}C_{B.albyi}$, $\delta^{18}O_{L.friesii}$, BIT, and $\delta^{13}C_{C.ung}$), we used instead the L-
method (Lanzante, 1996), which is based on the Mann-Whitney U test. In STARS, we set the *p*-value at 0.05, Huber's
weight parameter of 1, and the window size at 10. The selection of window size is generally arbitrary and can strongly affect
the sensitivity of the algorithm. For this reason, we confirmed that the identified regime shifts were also detected when using
window size values from 5 to 20.

We investigated the existence of size-dependent patterns in the $\delta^{18}O_{oto}$ and $\delta^{13}C_{oto}$ values of the otoliths of the two species by
testing for statistical correlation between the otolith size and the $\delta^{18}O_{oto}$ and $\delta^{13}C_{oto}$ values, since otolith size is linked to fish
size through species-specific functions (Edelist, 2014). For these tests, we used the Spearman rank correlation coefficient
(95% confidence level).

Furthermore, we examined possible covariance between the $\delta^{18}O$ and $\delta^{13}C$ values of the fish otoliths and foraminifera
(reflecting the responses of these marine organisms), and the paleoenvironmental variables, specifically the SST and SSS
values (Kontakiotis et al., 2022) and the BIT index (Butiseacă et al., 2022). For these tests, we used the Spearman rank
correlation coefficient (95% confidence level). We conducted these analyses across the entire study interval of the Agios
Myron section. Considering the observed lithological cyclicity, we repeated the analyses for the sapropel levels only.
Additionally, we used the results of the regime shifts detection, and we repeated the tests once more within each identified
regime. The length of the time series does not permit searching for lagged responses.

The above analyses were performed in R (version 4.3.1) (R Development Core Team, 2023), and we used the packages *dplyr*
(Wickham et al., 2023) and *rshift* (Room et al., 2020).

## 3 Results

### 3.1 Isotopic analysis results and size correlations

The $\delta^{18}O_{B.albyi}$ values range from −0.28 to 0.41‰ in the homogeneous marls, from −1.47 to 2.64‰ in the laminated marls,
and from −0.63 to 0.05‰ in the silty marls in the lower part of the section. The $\delta^{18}O_{G.obliquus}$ values range from –0.84 to
0.44‰ in the homogeneous marls, from –1.36 to 0.79‰ in the laminated marls, and from –0.74 to 0.23‰ in the silty marls.

The $\delta^{18}O_{L.friesii}$ values range from 2.14 to 4.30‰ in the homogeneous marls, from 2.32 to 4.11‰ in the laminated marls, and from 2.07 to 2.2‰ in the silty marls (Agiadi et al., 2024).

The $\delta^{13}C_{L.friesii}$ values range from −5.80 to −4.20‰ in the homogeneous marls, from −5.81 to −4.19‰ in the laminated marls, and from −5.41 to −4.68‰ in the silty marls in the lower part of the section. The $\delta^{13}C_{G.obliquus}$ values range from −0.18 to 0.46‰ in the homogeneous marls, from −0.35 to 1.49‰ in the laminated marls, and from 0.29 to 1.13‰ in the silty marls. The $\delta^{13}C_{B.albyi}$ values range from −5.92 to −5.23‰ in the homogeneous marls, from −9.02 to −3.71‰ in the laminated marls, and from −5.44 to −5.19‰ in the silty marls (Agiadi et al., 2024). There were no *Lesueurigobius friesii* otoliths in the

samples younger than 6.653 Ma.

No correlation was detected between the isotopic values and otolith length at the 95% confidence level, with Spearman's rho values of 0.13 (p = 0.58) for $\delta^{13}C_{B.albyi}$, 0.11 (p = 0.70) for $\delta^{13}C_{L.friesii}$, −0.04 (p = 0.88) for $\delta^{18}O_{B.albyi}$ and −0.17 (p = 0.56) for $\delta^{18}O_{L.friesii}$.

For the entire Agios Myron record, there is a strong correlation between $\delta^{18}O_{G.obliquus}$ and SSS (N = 24, rho = 0.78, p =

1.078e-05), but no correlation between $\delta^{18}O_{B.albyi}$ and SSS (N = 20, rho = 0.12, p = 0.61) or $\delta^{18}O_{C.ung}$ and SSS values (N = 22, rho = 0.30, p = 0.17). However, we found moderate correlation between $\delta^{18}O_{B.albyi}$ and $\delta^{18}O_{L.friesii}$ values (N = 10, rho = 0.69, p = 0.04).

Examining the laminated marls only, we found moderate correlation between $\delta^{18}O_{B.albyi}$ and $\delta^{18}O_{G.obliquus}$ (N = 20, rho = 0.55, p = 0.05), and $\delta^{18}O_{G.obliquus}$ and SST values (N = 24, rho = 0.60, p = 0.03). Strong correlation was found for the sapropel

levels between $\delta^{18}O_{G.obliquus}$ and SSS values (N = 14, rho = 0.93, p < 2.2e–16). No correlation was found between $\delta^{13}C_{B.albyi}$ and $\delta^{13}C_{G.obliquus}$ (N = 14, rho = 0.06, p = 0.84), $\delta^{18}O_{B.albyi}$ and SST (N = 14, rho = −0.09, p = 0.77), $\delta^{18}O_{C.ung}$ and SST (N = 14, rho = 0.24, p = 0.41), $\delta^{18}O_{B.albyi}$ and SSS (N = 14, rho = 0.39, p = 0.17), or $\delta^{18}O_{C.ung}$ and SSS values (N = 14, rho = 0.24, p = 0.41). Combining the laminated marls with the silty marl levels in the analysis, we found moderate correlations between $\delta^{18}O_{B.albyi}$ and $\delta^{18}O_{G.obliquus}$ (N = 16, rho = 0.50, p = 0.05), and $\delta^{18}O_{L.friesii}$ and $\delta^{18}O_{C.ung}$ values (N = 7, rho = 0.78, p = 0.05).

## 3.2 Regime shifts and responses

The Regime Shift Index (RSI) has positive values, which indicate the presence of regime shifts (95% confidence level), at: 1) 6.951 Ma for $\delta^{18}O_{G.obliquus}$ (RS1; RSI = 0.23), 2) 6.882 Ma for SST (RS2; RSI = 0.70), 3) 6.847 Ma for SSS (RS3; RSI = 0.85), and 4) 6.814 Ma for $\delta^{13}C_{G.obliquus}$ (RS4; RSI = 0.33). There is no correlation between any of the biogeochemical proxies used here from 7.166 to 6.951 Ma (RS1). After 6.951 Ma (RS1), very strong correlation was indeed found between

$\delta^{18}O_{G.obliquus}$ and SSS values (N = 15, rho = 0.85, p < 2.39e–05). Examining the laminated marls, there is only a weak correlation between $\delta^{18}O_{B.albyi}$ and $\delta^{18}O_{G.obliquus}$ values (N = 12, rho = 0.56, p = 0.06) after 6.951 Ma, and the very strong correlation between $\delta^{18}O_{G.obliquus}$ and SSS values (N = 12, rho = 0.88, p < 9.166e–05). From 7.166 to 6.882 Ma (RS2), a moderate correlation was found between $\delta^{18}O_{G.obliquus}$ and SST (N = 11, rho = –0.63, p = 0.04) but only a weak correlation between $\delta^{18}O_{G.obliquus}$ and SSS values (N = 11, rho = 0.58, p = 0.07). After 6.882 Ma (RS2), there is only very strong

correlation between $\delta^{18}O_{G.obliquus}$ and SSS values (N = 13, rho = 0.93, p < 2.2e–16).  From 7.166 until 6.847 Ma (RS3),

$\delta^{18}O_{G.obliquus}$ is strongly correlated with SST ($N = 12$, rho = –0.62, p = 0.03) but weakly correlated with SSS values ($N = 12$, rho = 0.54, p = 0.07), whereas after 6.847 Ma there is only very strong correlation between $\delta^{18}O_{G.obliquus}$ and SSS values ($N = 12$, rho = 0.94, p < 2.2e–16). From 7.166 until 6.814 Ma (RS4), a moderate correlation is found between $\delta^{18}O_{G.obliquus}$ and SSS values ($N = 14$, rho = 0.69, p = 0.006). After 6.814 Ma (RS4), the correlation between $\delta^{18}O_{G.obliquus}$ and SSS values ($N = 10$, rho = 0.94, p < 2.2e–16) becomes very strong.

## 4 Discussion

### 4.1 Paleoceanographic conditions

Our results evidence a significant correlation between otolith and foraminifera $\delta^{18}O$ values at surface and bottom waters, collected from the laminated and the silty marls at the base of the section. This result confirms $\delta^{18}O_{oto}$ as a promising proxy for paleoceanographic studies (as anticipated by e.g. Radtke et al., 1996; Thorrold et al., 1997; Darnaude et al., 2014), supporting the combined use of $\delta^{18}O$ measured on both otoliths and foraminifera to reveal changes in the life history and migration patterns of teleost fishes in deep time.

As expected, the $\delta^{18}O$ of the bottom-dwellers ($\delta^{18}O_{L.friesii}$ and $\delta^{18}O_{C.ung}$) have more positive values than those of the surface-dwellers ($\delta^{18}O_{B.albyi}$ and $\delta^{18}O_{G.obliquus}$), the benthic fish and foraminifera occupy the bottom of the water column, which is colder and more saline than the surface water (Fig. 2).

Sea surface salinity fluctuations in the eastern Mediterranean were the dominant factor controlling the biomineral isotopic composition of zooplankton and fishes already from 6.882 Ma (RS2) and until 6.5 Ma, as seen by the $\delta^{18}O_{G.obliquus}$–SSS and $\delta^{18}O_{B.albyi}$–SSS correlations. Between 7.166 Ma and 6.951 (RS1), surface conditions at Agios Myron were mostly controlled by SST, as shown by the correlation between $\delta^{18}O_{G.obliquus}$ and SST. For the interval between 6.951 (RS1) and 6.882 Ma (RS2), both SSS and SST drive the zooplankton and fish responses.

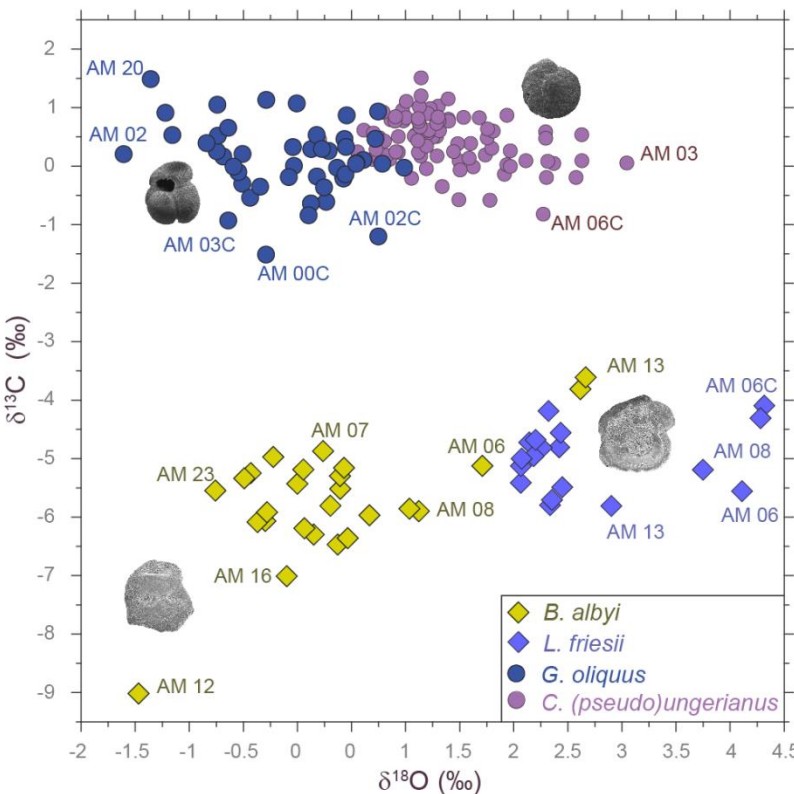

**Figure 2: $\delta^{18}$O–$\delta^{13}$C plot of the fish otolith and foraminifer samples analyzed from Agios Myron section. The four species clusters are clearly distinguished in the plot. Labels of several atypical samples are displayed.**

The regime shift of $\delta^{13}$C$_{G.obliquus}$ values at 6.814 Ma (RS4; Fig. 3) takes place in co-incidence with a high-amplitude disturbance of $\delta^{18}$O$_{B.albyi}$, mimicking the change in the Archaea community (isoGDGT2/isoGDGT3 index). Considering that $\delta^{18}$O$_{L.friesii}$ does not show a similar disturbance, we interpret the $\delta^{18}$O$_{B.albyi}$ pattern to reflect the short-lived event of an intermittent remarkable negative water budget at 6.824 Ma (Butiseacă et al., 2022). Subsequently, from 6.814 Ma until 6.5 Ma, $\delta^{18}$O$_{L.friesii}$ and $\delta^{18}$O$_{C.ung}$ values exhibit increasing trends, in contrast to $\delta^{18}$O$_{B.albyi}$ and $\delta^{18}$O$_{G.obliquus}$, which suggests that the event at 6.824 Ma triggered persistent stratification and increased bottom-water salinity. This interpretation is also corroborated by the increase in abundance of Neogloboquadrinids (Zachariasse et al., 2021) that are indicators of high nutrient availability in a colder, stratified water column (Sierro et al., 2003). The presence of high-salinity–tolerant benthic foraminifera and concomitant Mn/Al depletion at this time interval suggest decreased bottom-water oxygenation throughout the Mediterranean after a gateway restriction step at 6.8 Ma (Kouwenhoven et al., 2003; Sierro et al., 2003; Lyu et al., 2022). Since diagenesis has been shown to affect in a similar way the otoliths of both pelagic and benthic fishes, at least in a continental shelf environment (Agiadi et al., 2022), this benthic–pelagic decoupling after 6.824 Ma is considered to indeed reflect paleoenvironmental change. In contrast, a small, positive shift of $\delta^{18}$O$_{B.albyi}$ is observed at 6.68 Ma, which co-occurs

with a positive shift of $\delta^{18}O_{L.friesii}$, possibly reflecting similar changes in both surface and bottom waters, most likely due to an SST drop to 22 °C rather than SSS decrease that would bring the $\delta^{18}O_{oto}$ towards lower values (Kontakiotis et al., 2022).

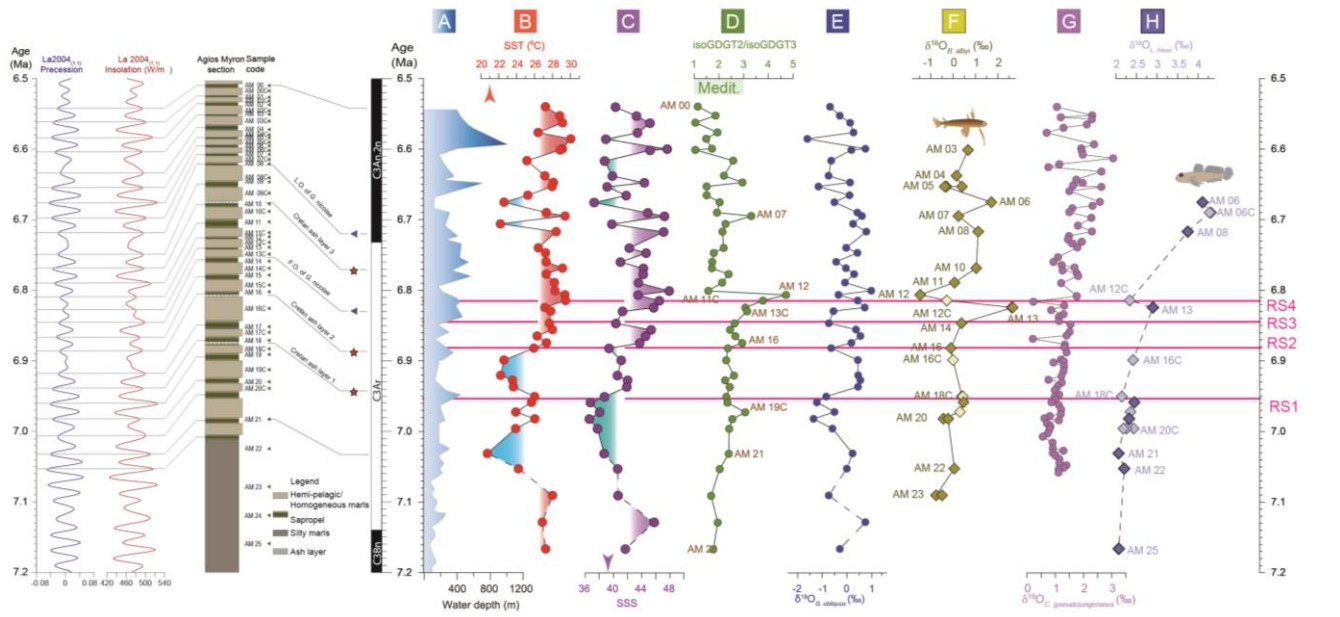


**Figure 3:** **Agios Myron lithostratigraphic column, major bioevents and chronostratigraphy, in the precession and insolation framework for the time interval of the study (Laskar et al., 2004). A) paleobathymetric curve based on foraminifera (Zachariasse et al., 2021); B) sea surface temperature calculated based on TEX$^H_{86}$, C) sea surface salinities based on SST-TEX$^H_{86}$ and $\delta^{18}O_{G.obliquus}$ (Kontakiotis et al., 2022); D) Ratio of isoprenoidal glycerol dialkyl glycerol tetraethers 2 and 3 (isoGDGT2/isoGDGT3)**

**(Butiseacă et al., 2022); E) $\delta^{18}O$ of the shells of the planktonic foraminifer *Globigerinoides obliquus*; F) $\delta^{18}O$ of the otoliths of the pelagic fish *Bregmaceros albyi*; G) $\delta^{18}O$ of the shells of the benthic foraminifer *Cibicides (pseudo)ungerianus* (Zachariasse et al., 2021); and H) $\delta^{18}O$ of the otoliths of the benthic fish *Lesueurigobius friesii*. RS1–4 indicate the four regime shifts found in this study at 6.951 Ma for $\delta^{18}O_{G.obliquus}$, 6.882 Ma for SST, 6.847 Ma for SSS, and 6.814 Ma for $\delta^{13}C_{G.obliquus}$.**

### 4.2 Fish metabolic response to paleoceanographic change

Carbon in otoliths originates from two sources: seawater and diet. Therefore, $\delta^{13}C_{oto}$ values reflect changes in the seawater $\delta^{13}C$, which depends on pH, and in the fish's diet and metabolic rate. The $\delta^{13}C_{oto}$ values here are more negative than those of the foraminifera (Fig. 2), which is expected since $\delta^{13}C_{diet}$ is more negative than $\delta^{13}C_{DIC}$. Considering $\delta^{13}C_{G.obliquus}$ and $\delta^{13}C_{C.ung}$ values as reflecting $\delta^{13}C_{DIC}$ in surface and bottom waters, respectively (Mackensen and Schmiedl, 2019), we evaluate here the $\delta^{13}C_{oto}$ changes as indicative of changes in the metabolic responses of the two fish species. Considering the

foraminifera $\delta^{13}C$ as a proxy of seawater $\delta^{13}C$, the $\delta^{13}C_{oto}$ values are mostly controlled by fish metabolism (Fig. 4E and 4H). Notably, the $\delta^{13}C_{oto}$ and the $\Delta^{13}C$ records (expressed as $\Delta^{13}C_{G.obliquus-B.albyi}$ and $\Delta^{13}C_{C.ung-L.friesii}$) exhibit the same trends both for surface (Fig. 4E) and bottom waters (Fig. 4H).

For fish feeding on phytoplankton and zooplankton, such as the ones examined in this study, $\delta^{13}C_{oto}$ is negatively correlated with food availability and, therefore, with net primary productivity (Burton et al., 2011): under conditions of high (low) food availability, individual fishes with high (low) metabolic rates are more resilient and they would exhibit lower (higher) $\delta^{13}C_{oto}$ values since they would be consuming more, consequently acquiring more $^{12}C$ from their diet. In addition, $\delta^{13}C_{oto}$ values may be positively, negatively, or not correlated at all with temperature depending on the species, oceanographic conditions and setting (Kalish, 1991; Martino et al., 2019). In the case of Agios Myron, no significant correlation is found between $\delta^{13}C_{B.albyi}$ (or $\delta^{18}O_{B.albyi}$ either), and SST or SSS values. Both $\delta^{13}C_{B.albyi}$ and $\delta^{18}O_{B.albyi}$ show much wider ranges in the laminated than in the homogenous marls, but these are not accompanied by greater ranges of SST or SSS in the sapropelitic levels (Kontakiotis et al., 2022). $\delta^{13}C_{B.albyi}$ alone could be explained by wider ranges in primary productivity and nutrient supply among the laminated marls. In that case, the $\delta^{18}O_{B.albyi}$ range would mean that the fishes also experienced larger ranges in salinities and/or temperatures (compared to the homogeneous marl levels). But such a hypothesis is not supported by the SST and SSS records (Fig. 3). Alternatively, the $\delta^{13}C_{B.albyi}$ and $\delta^{18}O_{B.albyi}$ ranges (Figs. 3 and 4) could be attributed to shifts in the depth distribution of *B. albyi* to include deeper, even anoxic parts of the water column, as is common for the modern *B. nectabanus* (FishBase, 2024).

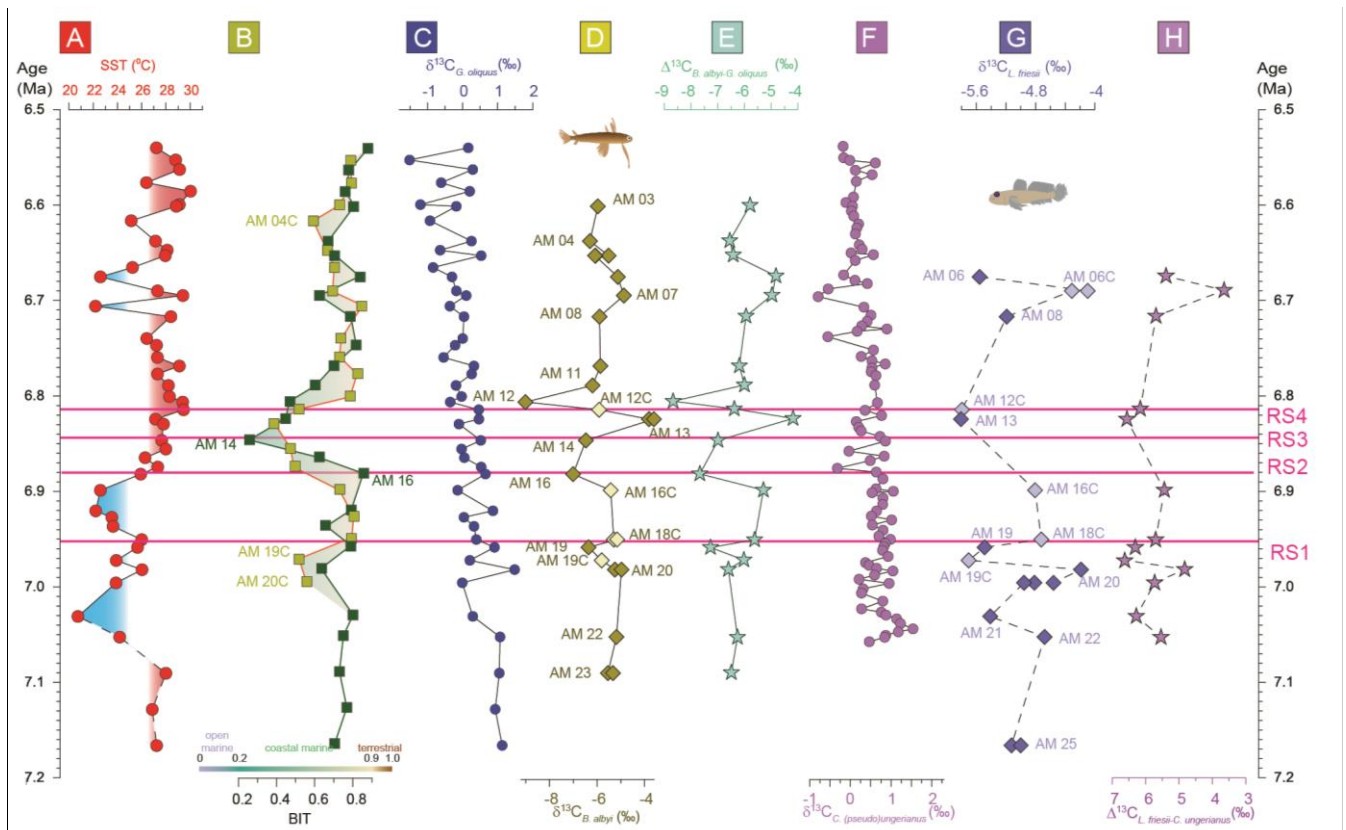

**Figure 4:** Agios Myron A) sea surface temperature calculated based on $TEX^{H}_{86}$ (Kontakiotis et al., 2022); B) Branched and Isoprenoid Tetraethers (BIT) index (Butiseacă et al., 2022); C) $\delta^{13}C$ of the shells of the planktonic foraminifer *Globigerinoides*

*obliquus*; D) $\delta^{13}$C of the otoliths of the pelagic fish *Bregmaceros albyi*; E) Difference between $\delta^{13}$C$_{B. albyi}$ and $\delta^{13}$C$_{G. obliquus}$; F) $\delta^{13}$C of the shells of the benthic foraminifer *Cibicides (pseudo)ungerianus* (Zachariasse et al., 2021); G) $\delta^{13}$C of the otoliths of the benthic fish *Lesueurigobius friesii*; H) Difference between $\delta^{13}$C$_{L.friesii}$ and $\delta^{13}$C$_{C.ung}$. RS1–4 indicate the four regime shifts found in this study at 6.951 Ma for $\delta^{18}$O$_{G.obliquus}$, 6.882 Ma for SST, 6.847 Ma for SSS, and 6.814 Ma for $\delta^{13}$C$_{G.obliquus}$.

The foraminifera $\delta^{13}$C records (Fig. 4C and 4F) follow the global decreasing trend. Both surface and bottom-water DIC become increasingly depleted in $^{13}$C from 7.2 to 6.5 Ma in the Agios Myron section (as both $\delta^{13}$C$_{G.obliquus}$ and $\delta^{13}$C$_{C.ung}$ show negative trends; Fig. 4). In the absence of a similar trend in the BIT index (i.e. a change in the terrestrial versus aquatic source of organic matter), these trends reflect decreased productivity and ventilation of bottom waters, in line with the North Atlantic and Western Mediterranean records (Hodell et al., 2001; van der Schee et al., 2016; Drury et al., 2018; Bulian et al.,

2023, 2022). In contrast, otoliths cannot be used as proxies for the $\delta^{13}$C of seawater, because they are strongly influenced by the $\delta^{13}$C of the fishes' diet, which varies greatly. For example, *Bregmaceros* spp. feed on zooplanktonic invertebrates (mostly copepods), but some species are also known to eat phytoplankton (FishBase, 2024). As the exact composition of the Late Miocene fish diet in Agios Myron cannot be deetermined, we cannot attribute the observed $\delta^{13}$C$_{oto}$ shifts to changes in the abundance of particular organisms (e.g., copepods). Ultimately, $\delta^{13}$C$_{oto}$ is a proxy of the average fish metabolism

(Wurster and Patterson, 2003; Solomon et al., 2006; Trueman et al., 2016; Chung et al., 2019a, b; Martino et al., 2020; Smoliński et al., 2021; Trueman et al., 2023; Jones et al., 2023). Both $\delta^{13}$C$_{B.albyi}$ and $\delta^{13}$C$_{L.friesii}$ show negative peaks at 6.959, 6.899 and 6.806 Ma, which coincide with changes in the composition of the Achaea community (isoGDGT2/isoGDGT3; Fig. 3) and fall around the times of peaks in the BIT index. The statistical testing did not show a correlation between $\delta^{13}$C$_{B.albyi}$, $\delta^{13}$C$_{L.friesii}$ , BIT and isoGDGT2/isoGDGT3, and the time interval of this study does not permit an investigation of

any lagged responses. Additionally, we consider it highly unlikely that diagenetic effects would produce co-incident peaks in the carbon isotopic ratio of fish otoliths and BIT: on one hand, the carbon isotopic composition of fish otoliths reflects mostly the ratio of their diet items (in this case phytoplankton and zooplankton), whereas BIT is the ratio of terrigenous/aquatic GDGTs that come predominately from terrestrial soils, peats, lakes and rivers (Hopmans et al., 2004; Peterse et al., 2012), or Thaumarchaeota (mostly sourced by marine archaeol picoplankton; Sinninghe Damsté et al., 2002).

Therefore, we consider that these $\delta^{13}$C$_{B.albyi}$ and $\delta^{13}$C$_{L.friesii}$ negative peaks reflect abrupt, regional events of crisis in the food availability (source and content) for surface and bottom-water fishes, which impacted fish metabolism.

A single regime shift in $\delta^{13}$C$_{G.obliquus}$ is found at 6.814 Ma (RS4) and it coincides with a change in the Archaea community (isoGDGT2/isoGDGT3 index; Fig. 4; Butiseacă et al., 2022). The event at 6.82–6.81 Ma has a strong impact on fish metabolism, as it is clearly seen first in the negative peak of $\delta^{13}$C$_{B.albyi}$ and then in the positive shifts of $\delta^{13}$C$_{B.albyi}$ and

$\delta^{13}$C$_{L.friesii}$, which are accompanied by a BIT shift to increased relative contribution of terrestrial organic matter (Butiseacă et al., 2022) and to the maximum of isoGDGT2/isoGDGT3. Together with the positive $\delta^{18}$O$_{B.albyi}$ shift at 6.824 Ma that is attributed to a negative water-budget event (see section 4.1), we propose a temporary influx of high-nutrient terrestrial material at 6.814 Ma, leading to phytoplankton bloom and a consequent increase in zooplankton abundance, which provides

more food for the fish population both in surface and bottom waters. After this point, $\delta^{13}C_{L.friesii}$ increases and peaks at 6.690 Ma (Fig. 4) indicating a corresponding decrease in metabolic rate, and then it abruptly drops at 6.675 Ma before the disappearance of *L. friesii* from Agios Myron and other Mediterranean sites (Karakitsios et al., 2017; Schwarzhans et al., 2020). At 6.690 Ma, there is a pronounced drop in $\delta^{13}C_{C.ung}$, but no corresponding change in $\delta^{13}C_{G.obliquus}$, $\delta^{18}O_{C.ung}$ or $\delta^{18}O_{G.obliquus}$, suggesting that decreased bottom-water ventilation, combined with basin deepening at the Agios Myron location (Zachariasse et al., 2021) resulted in the local extirpation of *L. friesii*.

Evolutionary adaptation is expected to reduce the effects of environmental change on physiology, but when multiple environmental stressors act contemporaneously, their interactions largely determine the net effect (Petitjean et al., 2019). In the early Messinian, temperature and salinity in the Mediterranean Sea (Vasiliev et al., 2019; Kontakiotis et al., 2019, 2022), as well as the source and amount of organic matter available (including from primary production; Kontakiotis et al., 2020; Butiseaca et al. 2022) showed large-amplitude variability. Temperature and salinity would be expected to act synergistically as they both increase metabolism (Clarke and Johnston, 1999; Gillooly et al., 2001; Komoroske et al., 2016), but the roles of primary productivity and organic matter type are unclear. At the evolutionary scale, to survive, the species would have to adapt to the prolonged and compounded effects of these multiple stressors, considering also that the frequency of shifts between warm/higher-salinity and cold/lower-salinity intervals increased after 6.72 Ma (Kontakiotis, Butiseaca et al., 2022). Our results suggest that the pelagic fish species indeed adapted well to the combined effects of temperature and salinity changes, while their benthic counterparts could not adapt after 6.814 Ma (RS4). After this tipping point, a change in the source and amount of organic matter also took place, which we believe led to the local maladaptation and finally extirpation of the benthic fish communities.

## 5 Conclusions

In the pre-MSC Messinian, due to the restriction of the Mediterranean–Atlantic gateway, the Mediterranean experienced intense variability in temperature and salinity, but also changes in primary productivity and terrestrial influx, which resulted in increased stratification of the water column and decreased ventilation of the bottom waters (Fig. 5). Our results evidence turning points in the Mediterranean paleoceanographic conditions between 6.951 and 6.882 Ma, when salinity and its variation amplitude increase probably related to stratification of the water column, and at 6.814 Ma when there is a major disturbance in the influx amount and source of organic material to the basin driving a change in primary production. For fishes, the most important event is the regime shift at 6.814 Ma (RS4), after which we see a pelagic–benthic decoupling with $\delta^{18}O_{L.friesii}$ expressing the higher salinity of the lower part of the water column, as well as a rapid increase and then drop in the metabolic rate of the benthic fish ($\delta^{13}C_{L.friesii}$), ultimately leading to the local extirpation of the species.

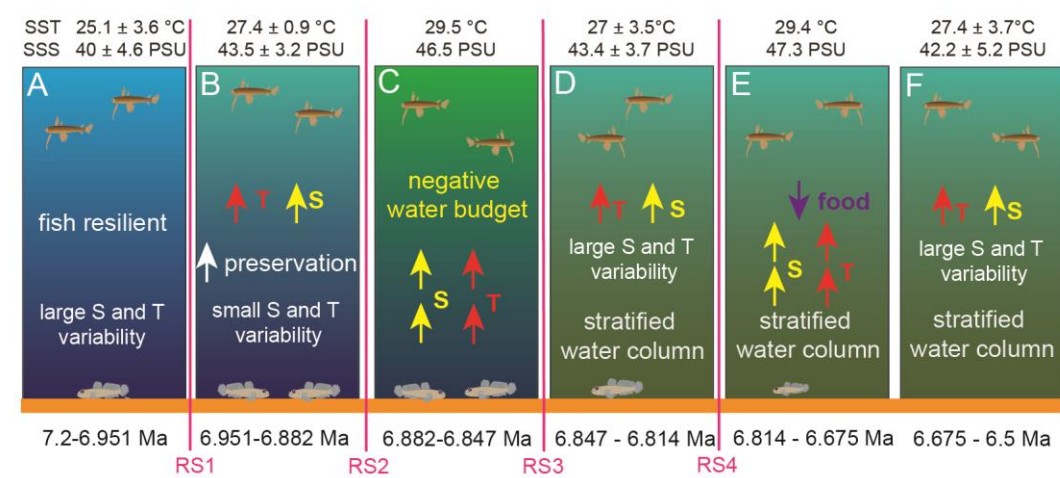

**Figure 5:** Schematic model of the response of surface and bottom-dwelling marine fishes to the paleoceanographic and paleoclimatic changes taking place from 7.2 to 6.5 Ma in the Eastern Mediterranean. RS1–4 indicate the four regime shifts found in this study at 6.951 Ma for $\delta^{18}O_{G.obliquus}$, 6.882 Ma for SST, 6.847 Ma for SSS, and 6.814 Ma for $\delta^{13}C_{G.obliquus}$.

## Data availability

Raw data for this manuscript are publicly available at: https://doi.org/10.5281/zenodo.10602427

## Author contributions

Conceptualization: KA, IV; Data curation: IV, GK; Formal analysis: KA; Funding acquisition: KA, IV; Investigation: KA, IV, GB, GK, DT, EB, SZ, EK, AA, AM; Methodology: KA; Visualization: KA, IV; Writing – original draft preparation: KA, IV; Writing – review & editing: GB, GK, DT, EB, SZ, EK, AA, AM.

## Competing interests

The authors declare that they have no conflict of interest.

## Acknowledgements

The authors would like to thank Prof. Clive Trueman for his insightful comments on an early version of this paper, as well as the Anonymous Reviewers for their very constructive comments. This research was funded in part by the Austrian Science Fund (FWF) [grant DOI 10.55776/V986; project "Late Miocene Mediterranean Marine Ecosystem Crisis" (2022–2026)] (KA). For open access purposes, the author has applied a CC BY public copyright license to any author accepted manuscript

version arising from this submission. This work was also supported by Greek national funds and the European Social Fund through the action "Postdoctoral Research Fellowships" of the Greek National Scholarships Foundation, project "Comparative study of the Messinian Salinity Crisis effect on the Eastern Ionian and northern Aegean ichthyofauna" (2017–2019) (KA); and by the Greek-German collaboration project (IKYDA-DAAD): "Quantification of the environmental changes in the Eastern Mediterranean at the onset of the Messinian Salinity Crisis (Crete-Greece)" (QUANTMES) (IV). Collaboration for this project was facilitated by the COST Action CA15103 "Uncovering the Mediterranean salt giant" MEDSALT (2016–2020).

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
