# Peer review of "Coupled otolith and foraminifera oxygen and carbon stable isotopes evidence paleoceanographic changes and fish metabolic responses"

_EGUsphere, 2024_

## Author Response (AR1)

**Response to Reviewers' comments**

**Reviewer 1**

The manuscript titled "Coupled Otolith and Foraminifera Oxygen and Carbon Stable Isotopes Evidence Paleoceanographic Changes and Fish Metabolic Responses" by Agiadi et al. presents a novel and significant contribution towards understanding the impact of extreme events, particularly the preconditioning stages of the Mediterranean Salinity Crisis, on marine fish life. This study stands out for its incorporation of higher organisms, adding substantial value to the research. As the authors state, the Mediterranean is indeed a great example of how different groups of organisms cope with changes in oxygen levels, nutrients and salinity and therefore I believe that this work is of extreme interest for many scientists dealing with the evolution of the Mediterranean, both from a biological and geological perspective. The manuscript is clear, well written and the obtained data are coherently interpreted.

Apart from some minor changes to be applied to the text some clarifications are needed:

Where does the age model come from? Has it been already published in previous works?

The age model for the Agios Myron section has been published by Zachariasse et al. (2021). We have added this information in the revised text. The Agios Myron section presents lithological precessional cycles (confirmed by the Vanadium concentrations in the bulk sediment and spectral analysis), which have been orbitally tuned through correlation with the Metochia section (Hilgen et al. 1995; 1997; Krijgsman et al. 1995; 1999), and anchoring in planktonic foraminifera bioevents and ash layers.

Elaborate or make more explicit why the 13C trend in otoliths does not reflect the Mediterranean or Global trend of decrease. Is it because the strong influence that nutrients have?, should then the trend be opposite in this specific case? Does it mean that they reflect more local changes and cannot be used for global changes in climate? Do they highlight punctual more abrupt changes?

The foraminifera $\delta^{13}$C curves express the global decreasing trend. In contrast, otoliths cannot be used as proxies of the carbon isotopic composition of seawater, because they are strongly influenced by the $\delta^{13}$C of the fishes' diet items, which vary greatly. For example, Bregmaceros spp. feed on zooplanktonic invertebrates (mostly copepods), but some species are also known to eat phytoplankton. Because we cannot know which was the exact composition of the diet of each individual fish in the Late Miocene Agios Myron area, the $\delta^{13}$C of those diet items or the incorporation ratio to otoliths, we cannot attribute the otolith $\delta^{13}$C shifts to a change in the abundance of particular organisms (e.g. the copepods). Ultimately, otolith $\delta^{13}$C is a proxy of fish metabolism, their capacity to grow, as has been demonstrated by previous studies (e.g. Wurster and Patterson, 2003; Solomon et al., 2006; Trueman et al., 2016; Chung et al., 2019a, b; Martino et al., 2020; Smoliński et al., 2021; Trueman et al., 2023; Jones et al., 2023). In the context of our study, we consider that the observed $\delta^{13}$C reflect shifts in the metabolism of the particular fish species in the region of study, not globally. The coincidence of these with the BIT and

isoGDGT2/isoGDGT3 shift from the same area suggests that these indeed are abrupt.

Line 113: parenthesis missing for reference (Whitley, 1941)

The original description of *Bregmaceros nectabanus* was by Whitley (1941), so no parenthesis is needed,

https://www.marinespecies.org/aphia.php?p=taxdetails&id=217741

Line 115: has been reported

corrected

Line 119: Today, it has a preferred temperature….

corrected

Line 244: Strange sentence, not clear, n that it requires substitute

We replaced the sentence with:

"In that case, the $\delta^{18}O_{B.albyi}$ patterns would mean that the fishes also experienced larger ranges in salinities and/or temperatures (compared to the homogeneous marl levels). But these are not supported by the SST and SSS curves (Fig. 3)."

**Reviewer 2**

This work is novel as it attempts to explore physiological responses of higher trophic level organisms (fishes) to large scale climatic fluctuations. Generally in deep time studies, physiological responses are inferred (e.g. as a response to temperature or via changes in body size or species compositions). Direct reconstruction of organism physiology across time series of environmental change is an exciting development. I would comment in general that co-incidence of physiological change with temperature / salinity change does demonstrate causation and likely cannot be used to unequivocally identify drivers of local extirpation– but it is certainly interesting and informative to explore physiological responses.

Another important angle to the work which is not developed in the introduction is the perspective that deep time studies can bring to modern ecological-evolutionary studies – for instance evolutionary adaptation within populations is expected to reduce long term physiological effects of drivers (e.g. temperature) at least within the organisms' thermal window. Deep time studies provide ability to test the capacity for species and communities to resist physiological effects of directed environmental change – this could be brought out a bit more in the intro / conclusion.

We believe this study will form the basis for further works investigating the long-term effects of multiple environmental stressors on fishes. Notably, the Mediterranean, particularly during the Late Miocene, offers great opportunities to explore the

evolutionary responses of marine organisms to environemtnal changes, which are indeed very relevant to modern ecological studies.

The idea to consider our results in terms of the species' resilience is very intriguing. Thank you for this!! We added the following passage in the Introduction:

"The modern trajectory of climate change makes the need for an improved understanding of the long-term resilience of marine fishes and their populations urgent. Over time, in the face of environmental change, the response of organisms shifts from phenotypic plasticity, to physiological changes, to evolutionary adaptation, or otherwise to extinction (Donelson et al., 2019). To this end, deep-time studies can contribute greatly, because they provide the necessary baselines, past analogues and long time series (Dietl et al., 2015; Leonhard and Agiadi, 2023)."

For the discussion, we explained that indeed evolutionary adaptation could reduce the effects on physiology, but this becomes more complicated because of the potential interactions between the multiple environmental stressors that affected fishes in this case (Petitjean et al., 2019). In the Early Messinian Mediterranean, temperature and salinity both showed large-amplitude variability (Vasiliev et al., 2019; Kontakiotis et al., 2022), and the source and amount of organic matter available (including from primary production) varied as well (Butiseaca et al. 2022). Temperature and salinity would be expected to act synergystically as they both increase metabolism (Clarke and Johnston, 1999; Gillooly et al., 2001; Komoroske et al., 2016), but the roles of primary productivity and organic matter type are unclear.

At the evolutionary scale, to survive, the species would have to adapt well to the prolonged and compounded effects of these multiple stressors, considering also that the frequency of the shifts between warm/salty and cold/fresh intervals increased after 6.72 Ma (Kontakiotis, Butiseaca et al., 2022). Our results suggest that the pelagic fish species indeed adapted well to the combined effects of temperature and salinity changes, but the benthic could not adapt after 6.814 Ma (Regime Shift 4). After this tipping point, also a change in the source and amount of organic matter took place, which we believe led to the local maladaptation and finally extirpation of the benthic fish.

I do have some requests for clarification around the data analyses. I would find it helpful in the data analysis section to define the purpose / hypothesis behind each analysis. I understand the test for size-dependency on otolith d13C values, but the reasoning underpinning the correlation tests (lines 141-145) were less clear to me. There is temporal auto-correlation expected – presumably the analyses are essentially looking for the effect of temporal autocorrelation rather than an effect of (say d18O foram on d18O fish…). Perhaps because I didn't fully understand the rationale for these correlations, I found the description of their relative strength– which are not visualised anywhere- difficult to follow, or to grasp the significance or otherwise of different correlations. Could the whole analysis be analysed in a single model with lithology as a fixed effect  - and with some consideration of temporal structure?  In this way, fixed effects of taxon (and interactions) could be used to explore similarities or differences in their responses to either salinity or temperature.

The tests were aimed to detect potential covariance between the environmental parameters and the responses of organisms, as reflected in the isotope ratios. The length of the time series does not permit us to search for lagged responses. In order to conduct the analysis in a single model, we would have to use only those samples that came from the same horizons and all variables were measured. Because the otolith measurements for the two species do not always come from the same horizons, removing the missing levels would greatly reduce the number of points to use in the analysis.

In order to clarify the hypothesis behind each test, we modified the text in lines 141–146 to read: «Furthermore, we examined possible covariance between the $\delta^{18}$O and $\delta^{13}$C values of the fish otoliths and foraminifera (reflecting the responses of these marine organisms), and the paleoenvironmental variables, specifically the SST and SSS values provided by Kontakiotis et al. (2022) and the BIT index (Butiseaca et al., 2022). For these tests, we used the Spearman rank correlation coefficient (95% confidence level). We conducted these analyses across the entire study interval of the Agios Myron section. Considering the observed lithological cyclicity, we repeated the analyses for the sapropel levels only. Additionally, we used the results of the regime shifts detection, and we repeated the tests once more within each identified regime. The length of the time series does not permit searching for lagged responses.»

It is difficult to determine if there are sufficient samples to validate the correlations in each parameterization.

We were able to obtain otoliths of *Bregmaceros albyi* from 20 stratigraphic horizons (out of the 47 sampled for foraminifera and the biomarker analyses) and of *Lesueurigobius friesii* from 14 horizons. The number of samples in each comparison is the smallest available of the two in the pair, as seen in the raw measurements available at https://doi.org/10.5281/zenodo.10602427. To facilitate evaluating this, we include the N for each comparison in the Results section of the revised manuscript.

I appreciate the novelty of attempting to recover metabolic responses to environmental drivers that forms the central argument to the study. The metabolism story of progressively increasing separation between d13Cforam and d13C otolith esp in the upper part of the sequence is really interesting, as that does imply that whatever causes the surface water DIC shift (argued for reducing surface productivity)is coincident with decreasing metabolic activity in fish – this is potentially a really nice observation. I am struggling a little to be certain that the shorter-term excursions, sometimes indicated by single data points can be reliably interpreted. The decoupling between pelagic and benthic fish time series is promising. Temporal co-incidence between d13Cshifts and other proxies is interesting, but I would at least want to see replication of the effect and to have things like diagenetic effects ruled out.

Thank you for this nice assessment of what we believe is indeed an important novelty in this work. Regarding the shorter-term excursions, we can only refer to these as co-occurrences/co-incidences. We are currently working to replicate this analysis in other areas in the Mediterranean. Neverthelss, we consider it highly unlikely that

diagenetic effects would produce co-incident peaks in the carbon isotopic ratio of fish otoliths and BIT: on one hand, the carbon isotopic composition of fish otoliths reflects mostly the ratio of their diet items (in this case phytoplankton and zooplankton), whereas BIT is the ratio of terrigenous/aquatic glycerol dialkyl glycerol tetraethers (GDGTs) that come from terrestrial soils, peats, lakes and rivers (Hopmans et al., 2004; Peterse et al., 2012), or Thaumarchaeota (mostly sourced by marine archaeol picoplankton; Sinninghe Damsté et al., 2002). Additionally, diagenesis would be expected to have the same effect on both pelagic and benthic fish otoliths (Agiadi et al., 2022), and therefore the decoupling between pelagic and benthic fish time series is also considered robust. We added explanations for these in the Discussion.

Line 60 – final sentence here doesn't follow – respiratory (metabolic) C has lower d13C value than DIC because of preferential incorporation of 12C during photosynthetic fixation of C at the base of the food chain.

Thank you for highlighting this: the «As a result» is deleted and the sentence is amended accordingly.

Materials and methods. A little more detail of exactly which horizons were sampled is needed. What mass / volume of sediment was sampled? Are these from the marl or sapropel (or both) horizons? Was any chemical used to break down sediment for wet sieving?

The horizons where each sample was taken from, along with their lithology, position along the section, age, and sample number have been provided in the raw data attached to this manuscript. The name and position in the section are also visible in Figure 3E. We sampled approximately 1 kg of sediment from each horizon, both sapropels and calcareous marls and we used plain water to break down the sediment. We have added this information in the Methods section of the revised manuscript.

L80 – and elsewhere  - numbers of replicates is potentially an issue here – why only one otolith per horizon?

For both otoliths and foraminifera, we analyzed replicates of 10% of the data to confirm reproducibility of the results. Specifically for otoliths, we analyzed replicates from horizons AM05 and AM20C. We included this information in the Methods section.

L83 – how was diagenetic condition assessed? Are the otoliths still purely aragonitic? Is there any secondary calcite? Line 93 clarifies that S microscopy was used, ok, but what do you expect for a well vs poorly preserved otolith? I guess this is much more poorly understood than for forams.

Actually, there have been some important studies on the diagenetic alteration of fossil fish otoliths (Dufour et al., 2000; Cook et al., 2015), which we now cite in the revised manuscript. Generally, otoliths seem to suffer little from diagenetic alteration, mostly maintaining their microstructure, chemistry and mineralogy. In fact, Dufour et al. (2000) discovered in their study of Pliocene otoliths from the Mediterranean that most specimens remained aragonitc, while pyrite appeared in only a few. Moreover,

the isotopic composition of otoliths depends on the taxon rather than geologic age (Dufour et al., 2000). Indeed, any diagenetic alteration is focused mostly on the outer part of the otolith (Cook et al., 2015). As is now standard in the field (e.g. Lopes et al., 2022), we excluded these cases by microscopic observation of the specimens prior to analysis.

L85 – fish taxa selected due to their 'well established ecology' – please summarise what is known… OK provided in line 119 (maybe move up as this is prior knowledge rather than newly inferred so doesn't need to be among the methods)

We moved this subsection 2.4 to before the oxygen and carbon analyses methodology in the revised manuscript.

L88-89 – I'm not clear what is meant by species specific differences in vital effects, or why selecting adult samples would mitigate against this. Indeed the premise of using d13C as a metabolic proxy is surely to quantify species specific differences in 'vital effects' (vital effects being a rather undefined geochemist term usually reflecting processes associated with metabolic rate…)

Indeed, $\delta^{13}C_{oto}$ reflects metabolism, but this precaution was also meant for $\delta^{18}O_{oto}$ and was based on the suggestions from previous studies (Darnaude et al., 2014). We modified this sentence now accordingly.

Line 95-100 Clarify - were whole otoliths crushed and homogenized, or was the outer surface analysed? This is critical to establish. Assuming whole otoliths were crushed and sub-sampled, d13C and d18O values reflect whole life conditions, likely averaged towards early life stages where otolith growth is fastest.

The otoliths were used whole, crushed and homogenized, and the results have been interpreted to reflect whole life conditions. This would indeed be a problem if we were dealing with species that drastically changed lifestyle and trophic level from young to adult. However, this is not the case for *Bregmaceros* or *Lesueurigobius* species (FishBase, 2024).

What mass of sample was analysed? Presumably gas was evolved with phosphoric acid (temperature and reaction time?)

We added the following information in the Methods section of the revised manuscript:

Each sample (between ~80 and ~110 µg) was weighed and transferred in a 12-ml glass vial. The airtight glass vials were placed into a Jumo iTRON 16 temperature controller of the A200S autosampler of the Finnigan™ GasBench II from Thermo Scientific, which is attached to the mass spectrometer. The autosampler allows automated isotope ratio determination of small $CO_2$ gas samples. To displace the atmospheric gas in the glass vials, each sample was flushed with 1.2 bar He (purity > 99.999 %) for 300 seconds. After this flush-fill viscous water-free 98 g/mol orthophosphoric acid ($H_3PO_4$) was injected with a syringe through the septum into each vial. $CO_2$ gas, phosphates of calcium and water will be formed by carbonate reacting with the acid. The vials were heated the temperature controller to a constant temperature of 72 °C to speed up the reaction between the carbonates and $H_3PO_4$,

and to prevent the crystallization of the acid at room temperature. For our samples, having high carbonate concentration, ten drops of $H_3PO_4$ were injected. The acid had a minimum reaction time of 45 minutes.

Line 100 (per mille not per mil)

Corrected

Throughout, please use d1O 'values' and d13C 'values'..

Done

Line 200 – careful with language here – text implies a 'response of zooplankton and fishes' to salinity or temperature – but actually you show a response of biomineral d18O values – which implies that the animals are not responding to temperature /salinity (i.e. they remain in place and their minerals record the fluctuations in the water chemistry).

We corrected the expression in the revised manuscript.

Line 205-215. I am a little uncomfortable with interpretations based on very small numbers of fish otolith data. For instance the shift in d18Ob.albini around 6.814 (if I understand the sampling correctly) is based on a single point / analysis?

We detected a regime shift at 6.814 Ma for $\delta^{13}C_{G.obliquus}$ using the STARS algorithm (Rodionov, 2004), which is capable of detecting regime shifts robustly even at the ends of a time series. Based on this result, in these lines, we discuss the patterns shown by the other variables before and after this regime shift (including a positive shift of $\delta^{18}O_{B.albyi}$ at 6.824 Ma). We revised this sentence in the revised ms.

Line 230 – I am happy with the idea of using d13Cotolith as a proxy for metabolic responses if the d13C of seawater is controlled. Why not report the difference between d13Cotolith and d13Cforam for each sample?

We added this in the revised manuscript in the revised Figure 4. The trends for both surface and bottom-water fish remain the same.

References

Clarke, A., Johnston, N.M., 1999. Scaling of metabolic rate with body mass and temperature in teleost fish. Journal of Animal Ecology 68, 893–905. https://doi.org/10.1046/j.1365-2656.1999.00337.x.

Cook, P. K. et al. 2015. Biogenic and diagenetic indicators in archaeological and modern otoliths: Potential and limits of high definition synchrotron micro-XRF elemental mapping. Chem. Geol. 414, 1–15. https://doi.org/10.1016/j.chemgeo.2015.08.017.

Dufour, E., Cappetta, H., Denis, A., Dauphin, Y., Mariotti, A., 2000. La diagenese des otolithes par la comparaison des donnees microstructurales, mineralogiques et

geochimiques: Application aux fossiles du Pliocene du Sud-Est de la France. Bull. Soc. Geol. France 171(5), 521–532.

Gillooly, J.F., Brown, J.H., West, G.B., Savage, V.M., Charnov, E.L., 2001. Effects of size and temperature on metabolic rate. Science 291(5538), 2248–2251.

Hilgen, F. J., Krijgsman, W., Langereis, C. G., Lourens, L. J., Santarelli, A., and Zachariasse, W. J., 1995. Extending the astronomical (polarity) time scale into the Miocene, Earth Planet. Sci. Lett., 136, 495–510, https://doi.org/10.1016/0012-821X(95)00207-S.

Hilgen, F. J., Krijgsman, W., and Wijbrans, J. R., 1997. Direct comparison of astronomical and 40Ar/39Ar ages of ash beds: Potential implications for the age of mineral dating standards, Geophys. Res. Lett., 24, 2043–2046, https://doi.org/10.1029/97GL02029.

Hopmans, E.C., Weijers, J.W.H., Schefuss, E., Herfort, L., Sinninghe Damsté, J.S., Schouten, S., 2004. A novel proxy for terrestrial organic matter in sediments based on branched and isoprenoid tetraether lipids. Earth Planet. Sci. Lett. 224, 107–116. https://doi.org/10.1016/j.epsl.2004.05.012.

Komoroske, L.M., Jeffries, K.M., Connon, R.E., Dexter, J., Hasenbein, M., Verhille, C., Fangue, N.A., 2016. Sublethal salinity stress contributes to habitat limitation in an endangered estuarine fish. Evol. Appl. 9, 963–981.

Krijgsman, W., Hilgen, F. J., Langereis, C. G., Santarelli, A., and Zachariasse, W. J., 1995. Late Miocene magnetostratigraphy, biostratigraphy and cyclostratigraphy in the Mediterranean, Earth Planet. Sci. Lett., 136, 475–494, https://doi.org/10.1016/0012-821X(95)00206-R.

Petitjean, Q., Jean, S., Gandar, A., Côte, J., Laffaille, P., Jacquin, L., 2019. Stress responses in fish: from molecular to evolutionary processes. Sci Total Env 684, 371–380.

Samor Lopes M, Dufour E, Sabadini-Santos E, et al. 2022. Stable isotopic analysis and radiocarbon dating of micropogonias furnieri otoliths (Sciaenidae) from southeastern Brazilian coast: seasonal palaeoenvironmental insight. Radiocarbon 64(5), 1109–1137. https://doi.org/10.1017/RDC.2022.57

Peterse, F., van der Meer, J., Schouten, S., Weijers, J.W.H., Fierer, N., Jacckson, R.B., Kim, J.-H., Sinninghe Damsté, J.S., 2012. Revised calibration of the MBT-CBT paleotemperature proxy based on branched tetraether membrane lipids in surface soils. Geochim. Cosmochim. Acta 96, 215–229. https://doi.org/10.1016/j.gca.2012.08.011.

Sinninghe Damsté, J.S., Schouten, S., Hopmans, E.C., van Duin, A.C.T., Geenevasen, J.A. J., 2002. Crenarchaeol: the characteristic core glycerol dibiphytanyl glycerol tetraether membrane lipid of cosmopolitan pelagic crenarchaeota. J. Lipid Res. 43 (10), 1641–1651. https://doi.org/10.1194/jlr.m200148-jlr200.